# 'We have a plan for that': a qualitative study of health system resilience through the perspective of health workers managing antenatal and childbirth services during floods in Cambodia

Dell D Saulnier [1], Dawin Thol,[2] Ir Por,[2] Claudia Hanson [1,3]
Johan von Schreeb,[1] Helle Mölsted Alvesson[1]

¹Department of Global Public Health, Karolinska Institutet, Stockholm, Sweden
²National Institute of Public Health, Phnom Penh, Cambodia
³Department of Disease Control, London School of Hygiene & Tropical Medicine, London, UK

**Correspondence to**
Dr Dell D Saulnier;
dell.saulnier@ki.se

## ABSTRACT

**Objective** Health system resilience can increase a system's ability to deal with shocks like floods. Studying health systems that currently exhibit the capacity for resilience when shocked could enhance our understanding about what generates and influences resilience. This study aimed to generate empirical knowledge on health system resilience by exploring how public antenatal and childbirth health services in Cambodia have absorbed, adapted or transformed in response to seasonal and occasional floods.

**Design** A qualitative study using semi-structured interviews and thematic analysis and informed by the Dimensions of Resilience Governance framework.

**Setting** Public sector healthcare facilities and health departments in two districts exposed to flooding.

**Participants** Twenty-three public sector health professionals with experience providing or managing antenatal and birth services during recent flooding.

**Results** The theme 'Collaboration across the system creates adaptability in the response' reflects how collaboration and social relationships among providers, staff and the community have delineated boundaries for actions and decisions for services during floods. Floods were perceived as having a modest impact on health services. Knowing the boundaries on decision-making and having preparation and response plans let staff prepare and respond in a flexible yet stable way. The theme was derived from ideas of (1) seasonal floods as a minor strain on the system compared with persistent, system-wide organisational stresses the system already experiences, (2) the ability of the health services to adjust and adapt flood plans, (3) a shared purpose and working process during floods, (4) engagement at the local level to fulfil a professional duty to the community, and (5) creating relationships between health system levels and the community to enable flood response.

**Conclusion** The capacity to absorb and adapt to floods was seen among the public sector services. Strategies that enhance stability and flexibility may foster the capacity for health system resilience.

## Strengths and limitations of this study

► Participants in a variety of roles and from multiple levels of the health system strengthened the diversity of perspectives on health service functioning during floods.
► Extensive discussion among the diverse research team following all interviews and a comprehensive audit trail increased the dependability and credibility of the study.
► The data collectors were familiar with rural Cambodia and health system and health services at multiple levels, which helped the quality of the interviews.
► Participants at higher levels of authority may have provided more socially desirable information or refrained from sharing the full extent of their experiences or thoughts.
► The participants' rationale for their actions when preparing and responding to floods was not always articulated, which may affect the reliability of the results.

## INTRODUCTION

Since early 2020, health systems worldwide have been severely challenged by the COVID-19 pandemic. Systems have faced a sudden demand to surveille, diagnose, and care for people with COVID-19 plus the continued demand for essential services like vaccination and maternal healthcare[1]; in some cases, the pandemic has already disrupted these services.[2] Attention has been increasingly placed on health system resilience as a way to reduce health system vulnerability and increase their ability to deal with shocks like the 2014–2016 West Africa Ebola outbreak, the COVID-19 pandemic or extreme weather events.[3–5]

Extreme weather events are projected to become more frequent and severe as a consequence of climate change, including an increase in floods.[6] By the end of the century, an additional 2.3 billion people are expected to be at risk of flooding.[7] The health impacts of floods are diverse, from direct mortality and morbidity to indirect poor health from their impact on societal functioning.[8 9] Health systems, crucial to managing the health of the population, are also directly and indirectly impacted by floods.[9–11]

Resilient health systems have the capacity to continue delivering health services when shocked or stressed, responding and adapting to the new health needs created by the shock or stress and to the routine health needs of the population.[3 12 13] Systems are exposed to shocks—sudden or extreme external phenomena that impact healthcare demand and health system resources—and to chronic, internal stresses that continually challenge the system's functioning, like persistent under funding.[14] Health systems that are exposed to severe shocks will need to undergo a greater degree of change to be resilient than systems that are exposed to less severe shocks and stresses.[15]

To date, the literature has focused on how to define and understand the concept of health system resilience; less has been published on how resilience can be generated.[5 14 16] Exploring what makes health systems resilient in real-world situations is necessary to understand what capacities could foster resilience to shocks like floods.

Cambodia is regularly exposed to repeated seasonal and occasional floods, as well as flood disasters. Improving health service delivery readiness to operate during floods has been an ongoing priority for the government of Cambodia since 2008,[17 18] and the health services have remained functional during previous flood disasters.[19] Concurrent with reforms that have focused on strengthening health service delivery and management, several strategic plans have been developed to enhance the health system's ability to respond to floods.[17 20 21] This study therefore aims to understand how health system resilience is generated by exploring the capacity of the Cambodian health system to absorb, adapt and transform during floods. We sought to understand how healthcare workers in public sector health facilities and health departments are able to continue providing health services during seasonal and occasional floods using antenatal and childbirth services as indicators of routine and new health needs, respectively.[22 23]

## Conceptual framework

The study was designed using the Dimensions of Resilience Governance framework.[24] The framework argues that a health system's ability to manage shocks and stress is dependent on the ability of the system and the people within it to manage four other capacities: (1) the ability to access, integrate and process *knowledge* from inside and outside the health system; (2) the ability to adapt in response by taking actions and decisions to anticipate and cope with *uncertainty*; (3) the capacity to manage interactions at multiple levels of the health system and with other systems (*interdependence*); and (4) the capacity to engage with users and communities to create a socially and contextually accepted system (*legitimacy*).[24] Health systems can then absorb shocks or stress using existing strategies and resources, adapt to them by temporarily adjusting how resources are used or transform the structure of the system in the long term to avoid them altogether.[5 25 26]

## METHODS

### Study setting

Cambodia is a predominantly rural, lower middle-income country. Rapid economic and developmental changes increased the life expectancy from 58.4 to 68.6 years between 2000 and 2015, and maternal health and service indicators have been improving over the last 20 years (table 1).[21 27] The country experiences annual, seasonal flooding along the main rivers and Tonlé Sap lake and occasional floods from monsoon rains in other regions. The National Committee for Disaster Management leads a programme for disaster management and flood response from the national to district level.[28] Operational preparedness and response plans exist for multiple levels

| Table 1  Maternal health and health service indicators in Cambodia[42–44] | | |
|---|---|---|
| | **2000** | **2017** |
| Maternal mortality ratio (deaths per 100 000 live births) | 488 | 160 |
| | **2000** | **2014** |
| Per cent of women receiving antenatal care at least once | 43.2 | 95.2 |
| Per cent of antenatal care received at a public facility | – | 94.0 |
| Per cent of childbirth care at a healthcare facility | 9.9 | 83.2 |
| Per cent of childbirth care at a public facility | – | 68.9 |
| Per cent of childbirth care at home | 89.0 | 16.6 |
| Per cent of childbirths attended by a doctor, nurse or midwife | 31.8 | 89.0 |
| Per cent of childbirths by caesarean section | 0.8 | 6.3 |

## Box 1  Existing preparedness and response measures for antenatal and childbirth services during floods[17]

The 2015 strategic plan for health during disasters prioritised building capacity for disaster risk reduction by strengthening governance and coordination between sectors and actors during emergencies, improving information and knowledge management and bolstering health service delivery and resources during disasters. Preparedness and response plans for health services during floods have been created for each province based on post-flood reviews and local knowledge of high-risk flood areas and the facilities within them. The plans include strategies such as stockpiling resources, surge capacity protocols and contingency budgets. Maternal health services were identified as an essential service at risk during floods, and specific strategies for antenatal and birth services include improving midwifery skills in general prior to flooding, providing medical kits with the relevant equipment to mobile outreach teams and ensuring pregnant women can be identified and evacuated during floods, if needed.

of the health system and include items such as roles and responsibilities during disasters (box 1).

The study used antenatal and childbirth services to explore service provision. These services are indicative of how health services for new and routine health needs function during floods, even though neither pregnancy nor childbirth or emergency pregnancy complications are causally linked to floods. Pregnancy is representative of routine health needs that do not change because of floods but still require preventive and promotive care that can be planned in advance, while deliveries and complications are representative of new health needs that can occur (sometimes unexpectedly) during floods and require skilled management and emergency care.[29 30]

In the Cambodian health system, districts link health centres and district hospitals to the provincial level through health departments, which oversee service delivery and provide operational support. During floods, public health facilities conduct additional health promotion, education and clinical outreach, and health departments prepare resources and oversee the flood response. Routine antenatal and uncomplicated childbirth care is provided at public health centres and private clinics. Additional antenatal services and emergency obstetric care is provided at private clinics and public hospitals.[31]

### Study design, participants and recruitment

This qualitative study used semi-structured interviews to explore public sector healthcare providers and health department staff experiences in delivering and managing antenatal health services during floods. Two districts were selected: one that typically floods every year (seasonal flooding) and one that experiences floods less often (occasional flooding) to cover the main frequencies of flooding in Cambodia.[29] The study included five public health centres serving a population of 10 000 each, two district referral hospitals serving populations of 200 000

people, two provincial-level and two district-level health departments, two communes and the Ministry of Health. The history of flooding in the provinces and districts was first discussed with the health departments to confirm recent flooding and obtain access to the health centres. The recently flooded health centres were then visited and enrolled after confirming that their catchment areas had flooded in the previous 5 months. Participants who had been working in their current role during the most recent flood were purposively selected for variety in their type of job, length of work experience, age and gender (table 2). All health centre and hospital chiefs were invited to participate and identified eligible midwives at their facilities. Health department and commune chiefs identified eligible participants among their staff, who were invited to participate.

### Data collection

Two female Cambodian data collectors (a midwife studying public health and a qualitative researcher and retired medical school lecturer) conducted and audio recorded 23 semi-structured interviews in Khmer. Interviews were conducted in November and December 2018 at the end of the rainy season to ensure that detailed experiences with recent flooding could be captured in the discussions. Participation was voluntary and unpaid, and all participants gave written or recorded verbal consent prior to the interviews. The interview guides (online supplemental file 1) were used to investigate participants' real-life experiences working with service provision and management during floods. Interview guide questions were developed from the four capacities in the Dimensions of Resilience Governance framework[24] which were adapted into concrete questions about the participants' experiences. Participants were interviewed individually in private areas such as empty meeting rooms. The second data collector had taught five of the participants in medical school since the 1980s; she acknowledged the connections and kept the interview focused on the interview guides. The quality of the interviews was assessed continually among the team through discussion and field notes. The 23 interviews were judged as sufficient to have captured the relevant information based on the information power concept, which states that samples that hold a high degree of relevant information require a smaller number of participants.[30] Interviews were transcribed verbatim into Khmer and translated into English by one translator. One participant requested that part of their interview be excluded from transcription and analysis. Uncertainties and inconsistencies between the transcripts and audio files during translation were discussed with the data collectors and the first author.

### Data analysis

A data-driven thematic analysis,[32] where codes are chosen based on a detailed analysis of the data rather than on

**Table 2** Characteristics of the study sites and participants

| Source | District | Summary of most recent flood | Professional area | Years in current job | Interview length (min) |
|---|---|---|---|---|---|
| Ministry of Health | – | – | Disaster management | 13 | 58 |
| Provincial health department | Seasonal | June to October; one-third of districts flooded | Maternal health | 15 | 57 |
| | | | Disaster management | 3 | 61 |
| Provincial health department | Occasional | July; half of districts flooded | Maternal health | 1 | 75 |
| | | | Disaster management | 3 | 64 |
| District health department | Seasonal | June to October; flooding around all health centres | Disaster management | 22 | 81 |
| District health department | Occasional | Unsure of dates; flooding around three health centres | Disaster management | 22 | 83 |
| Commune | Seasonal | July to October; in villages | Deputy commune chief | 4 | 61 |
| Commune | Occasional | August to October; in villages | Committee for Women's Affairs | 15 | 77 |
| Referral hospital | Seasonal | June to October; flooded villages, not near hospital | Hospital chief | 12 | 45 |
| | | | Midwife | 6 | 47 |
| Referral hospital | Occasional | July to August; flooded villages, not near hospital | Health centre chief | 19 | 36 |
| | | | Midwife | 4 | 45 |
| Health centre 1 | Seasonal | June to October; all villages in catchment flooded | Health centre chief | 6 months | 58 |
| | | | Midwife | 6 | 50 |
| Health centre 2 | Seasonal | August to October | Health centre chief | 1 | 64 |
| | | | Midwife | 28 | 61 |
| Health centre 3 | Occasional | September to November; half of villages in catchment area flooded | Health centre chief | 3 | 61 |
| | | | Midwife | 3 | 74 |
| Health centre 4 | Occasional | August to October; 2 of 7 villages in catchment area flooded | Health centre chief | 10 | 53 |
| | | | Midwife | 4 | 77 |
| Health centre 5 | Occasional | October to November; 2 of 7 villages in catchment area flooded | Health centre chief | 18 | 57 |
| | | | Midwife | 8 | 97 |

theory, was conducted in NVivo V.12 Pro. It was chosen to ensure that the theoretical concepts of the framework did not eclipse other relevant information in the data about the health system's functioning and to capture ideas that spanned multiple dimensions in the framework. Codes were developed by DDS that reflected descriptions or ideas about influences on health service delivery during floods. After coding was complete, the authors explored how the capacities of knowledge, uncertainty, interdependence and legitimacy from the framework could be influencing the relationship between codes. Categories were then created from the patterns of meaning from the data-driven coding plus the theoretical understanding gained during the previous step and a single theme was identified that captured the meaning and association between the categories. Lastly, the findings were mapped back onto the framework to illustrate how they contributed to the framework's capacities (figure 1).

**Patient and public involvement**

No patients involved.

**RESULTS**

We identified one theme, 'Collaboration across the system creates adaptability in the response', derived from five categories (table 3). Collaboration and social relationships appeared to create clear boundaries to decision-making around antenatal and birth care during floods. Both seasonal and occasional floods were discussed as strains rather than acute shocks or chronic stresses. With a firm understanding of the decision-making boundaries, providers and staff were able to prepare and respond to these floods in a flexible but stable manner, resulting in absorptive and adaptive capacity. Overarching health system and contextual changes appear to be fundamentally changing public sector service use and function

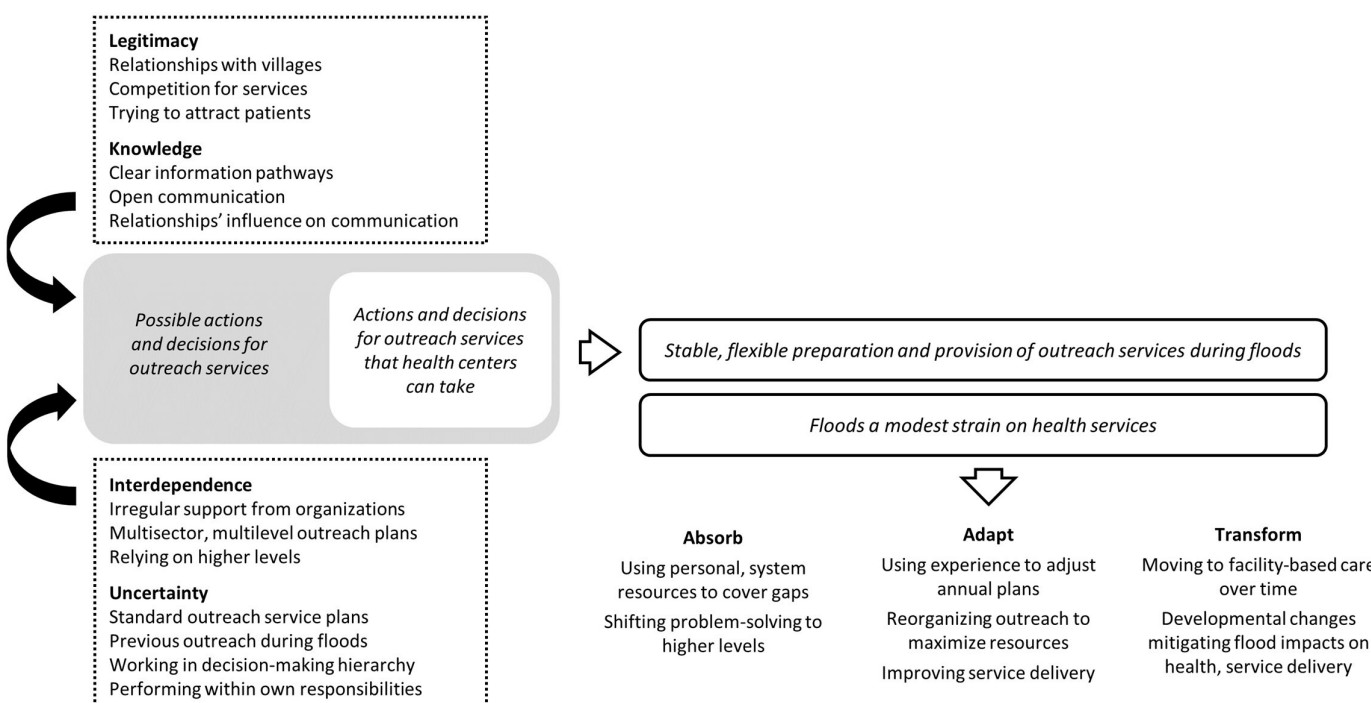

**Figure 1** A visual representation of the four dimensions of resilience and the system's capacity to absorb, adapt and transform as they relate to the main theme using health centre outreach services for antenatal and childbirth care during a flood as an example.

during floods, potentially transforming the system and the impact of floods themselves.

We illustrate the relationship between the theme and the Dimensions of Resilience Governance framework in figure 1, followed by a description of the categories informing the theme. The four dimensions feed into the process outlined in the theme, showing how boundaries on decision-making are contributing to preparation and response to modest seasonal and occasional floods, helping to create the capacity to absorb, adapt and transform.

**'Floods just another strain on services'**

Participants did not perceive seasonal or occasional floods as a threat to health service functioning. Instead, they were part of a larger constellation of constant pressures on service delivery. Providers and health department staff in both districts described pregnancy, childbirth and other health problems during floods as routine and 'nothing serious'. They note that antenatal and childbirth services usually functioned as normal during both types of floods. Changes over time, like better roads,

**Table 3** The theme, categories and subcategories describing the influences on public sector antenatal and childbirth health services during floods

| Theme | Categories | Subcategories |
|---|---|---|
| Collaboration across the system creates adaptability in the response | Floods just another strain on service delivery | |
| | Facilities and health departments able to calibrate and manoeuvre to make flood routines work for them | |
| | Working in the same direction during floods | |
| | Engaging in local governance to fulfil a duty to the community during floods | Health centres becoming obsolete during floods<br>Believing in the value of the system during floods |
| | Creating relationships to successfully respond to floods | Health centres as the linchpin for the health response to floods<br>Social relationships can challenge information sharing |

let participants feel that floods were no longer a major threat:

> The floods are not so big that we have to focus on them so much. They're not big to the point of an emergency. (Interview 7, provincial health department)

Participants described numerous, persistent challenges to service delivery that were independent of flooding. They included (1) slow, unreliable referral systems, (2) inadequate professional training, (3) heavy workloads and staff shortages, (4) unintegrated and undependable support from not-for-profit organisations, (5) poor or inaccurate health and system performance information, and (6) a slow and unresponsive process for requesting funds, resources and help from higher levels of the health system. Floods exacerbated these challenges but everyday strains on service provision subsumed the effect of floods:

> There's no shortage of medicine [during floods]. The hospital had bought and stored it at the hospital so that there are no shortages happening. It's just that there's no leftover money to encourage the staff. (Interview 20, referral hospital midwife)

### 'Facilities and health departments able to calibrate and maneuver to make flood routines work for them'

Staff at facilities and departments reviewed the standard flood response plans each year. Together with flood warnings and their experiences from previous floods, participants in both districts discussed how they knew the routines for preparing. They began to actively prepare in the months and weeks preceding the expected floods:

> All the plans are already written […] We have planned what to do, we have all events organised, and we just follow accordingly. We have written everything into the yearly plan, arranged it by month as well. And if there's a flood, we have listed down already what we need to do, what to buy, and so on. (Interview 2, health centre chief)

According to participants, facilities and health departments in both districts calibrated their plans and routines to fit different scenarios, and flexibly manoeuvred during flood response. They adapted the standardised plans—formulated for disasters—to fit large and small scale floods in each district. The adaptions were based on previous experience and scenarios they might encounter. The plans used strategies like: (1) diverting clinical staff between facilities and health departments to cover shortages or supplement outreach teams, (2) using discretionary budgets to prepare or respond, (3) piggybacking on the non-health response to floods, and (4) choosing how to organise outreach for flood-related care:

> We have a boat, it's even a motorboat. It costs more to transport the boat to the [flooded] area than it does to rent a local boat there. We ride a motorbike

and then ride on the boat with the locals. It costs less. (Interview 18, district health department)

### 'Working in the same direction during floods'

Facility chiefs and health department staff described a multilevel, multisectoral flood response in both districts. Bigger floods required a larger response and more organisation at the provincial level with external sectors. Participants knew that they could not control problems outside the health sector that affected health or their work during floods and needed to trust other sectors and higher authorities to find solutions:

> There're only the commune and the district that could solve the problem [of difficult travel during floods]. They [facility staff] can only complain to us but we can't do anything about it at all. For problems regarding the road, only the commune can make a plan and ask that the province supports it. (Interview 8, referral hospital midwife)

Participants noted that they had to collaborate between levels of the health system to keep facility-based services and clinical outreach functional during floods. They relied on higher organisational levels of the health system to be responsible for problem solving or to help with supplies, staff and expertise. They were free and able to contact higher levels for help, and described a strong sense of teamwork across levels when responding to floods:

> When there's a big flood, the district will have to visit the flooded areas every day to see the people's situation, how they are living, is there any clean water for them to use, any diarrhea, are there any health centre staff to help them with their health problems or not? Everyone is enthusiastic when there's a big flood, both the provincial level and district level. (Interview 10, district health department)

While free to ask for help, participants described clear limits to their decision-making. They stayed within the boundaries of their designated role when preparing and responding to floods. Their superiors made and approved most decisions about responding to bigger floods, noted in particular by midwives and health department staff:

> The decisions that I can make during a disaster without asking the upper level to help are none. During a disaster, you have to depend on the upper level. I have to report to them because we cannot work alone. (Interview 9, provincial health department)

### 'Creating relationships to successfully respond to floods'

Health department staff and providers were able to share information and keep facility-based care and outreach services functioning during floods, facilitated by their relationships with each other and the community.

## Health centres as the linchpin for the health response to floods

Participants saw the existing relationships between health centre staff and the community as crucial for providing health services during floods. Health centre staff worked to keep outreach and facility-based services functioning smoothly during floods. For example, they would pay out of pocket for gasoline or reimburse village leaders for emergency phone calls. Health centre staff felt that the community's knowledge of health risks and strategies to stay healthy during floods made it easier to provide services:

> The health centre doesn't have a boat, not even a tuk tuk [to transfer patients during floods]. The women need to have their own boats. The villagers are very smart. They know that they need to have a boat with them when they come for antenatal care. (Interview 14, health centre midwife)

Health centres and departments needed villages to supply information on their health and the severity of the floods in their area so that they could respond appropriately. According to participants, health centres needed to negotiate with commune and village leaders to allow health department staff to enter to villages during floods:

> Before we go to any village, we inform the health centre to inform [the local authorities]. If you don't inform them, they won't know what you're doing and there will be trouble. If it's before an election, you would have to be even more careful. (Interview 18, district health department)

## Social relationships can challenge information sharing

Participants described flood and health information as accessible to staff across all levels of the health system, structured to flow up or down from the village to the national level. Contact networks were openly accessed, regardless of rank or role. Open communication and information sharing meant contact was often between individuals rather than roles:

> Before they [the health centre] transfers a case here, they will call us. They call the hospital, there's a telephone there. Sometimes they contact [the hospital chief] directly. Sometimes they call any staff that they know here. (Interview 20, referral hospital midwife)

Personal and social relationships could hamper access and information sharing between individuals. Lower ranking staff in health departments and facilities, like midwives, only received information about how their facility would respond to floods on a 'need-to-know' basis. Participants explained that they were more willing to contact known, trusted individuals for information or help. Without trust, participants felt obligated to follow-up on information themselves to ensure it was correct:

> Sometimes, the information that is available is not clear. Then we have to go to the community for an assessment. Like in the past, we had information from the National Committee for Disaster Management that there were deaths [during the flood] but then when I went there, they [the health department] said there were none. So, it's hard. (Interview 23, Ministry of Health)

## 'Engaging in local governance to fulfil a duty to the community during floods'

Staff at the health centres remained engaged in holding themselves accountable for the care they delivered. They described feeling a professional duty and responsibility to the community's health during floods, despite feeling progressively less relevant in a changing landscape.

### Health centres becoming obsolete during floods

Participants described health centres as struggling to compete for services with private facilities and public hospitals regardless of floods. Health centre providers noted that more women now chose facilities with better services and resources and expressed difficulty understanding why they were losing relevance:

> It's not because there's a flood or not [that they go to the hospital to deliver]. We care for them since the first month to the eighth or ninth month, then they disappear. When it's an appointment date, they miss it. Then after four or five days, they carry the baby here. I ask where was the baby born? It was born at the public hospital. I ask why not at [my health centre]? They wanted to give birth there. That's their choice. (Interview 1, health centre midwife)

Health centre participants felt obligated to focus on facility-based services rather than outreach for flood-related care and felt tied to their facilities during floods. They had to prioritise staff, funds and equipment for the health centre, and felt less able to provide comprehensive antenatal outreach during floods:

> We have only one electric generator that is used at the health centre. So we cannot take that into the village [during outreach]. If we do, when people come to the health centre for treatment, then we can't provide full services to them. (Interview 4, health centre midwife)

As a result, clinical outreach during floods began when community members could no longer reach the health centres and focused on preventing communicable diseases.

### Believing in the value of the system during floods

During floods, health centres described attempts to increase demand at their facilities using schemes like waiving service fees and promoting facility-based care. They actively attempted to improve service quality at their facilities, for instance, by holding providers accountable for perceived poor treatment of patients.

Providers and health department staff felt pride in their work and a professional duty to serve the community, despite the persistent challenges described earlier. When faced with floods, they discussed sharing a common goal of helping the system work as it should, regardless of potential challenges. This echoed earlier expressions about relying on higher levels of the health system and the community to take responsibility:

> Cases of [women] unable to deliver, we never have them. We don't have any issue with it because we have the network to help her on time. If there are any problems, the village and commune chiefs have my phone number and the district head's phone number, if the patient needs an ambulance to take her from home. (Interview 19, hospital chief)

## DISCUSSION

This qualitative study highlights the importance of understanding the practices that enable health service provision during flooding. The results show an adjustable approach to preparing and responding to floods. The approach is grounded in collaboration and relationships across the system that set boundaries around actions and decisions, giving the system the capacity to adapt and absorb floods. This approach was similar between the seasonal and occasionally flooded districts, despite the difference in flood frequency. Health system strengthening initiatives and developmental changes in Cambodia, like reforms to promote and improve facility-based maternal health services and better roads, have transformed the health system context over time. The boundaries around actions and decisions, coupled with the transformative changes, had led to a system that is capable of maintaining pregnancy and childbirth services during floods, and one that views floods as strains rather than shocks.

The absorptive and adaptive capacities were characterised by the health facilities' ability to prepare and respond to seasonal and occasional floods in a stable but flexible way, with support from the health departments. Transformative capacity was not readily reported by the participants. Some changes in the structure or functioning of the system, such as the general shift to facility-based antenatal and childbirth care, were not explicitly linked to flooding. If the three capacities are seen as 'different perspectives of the same reality',[15] stability and flexibility are both necessary:[13 33] stability to cushion against shocks and let coordinated actions emerge, and flexibility to change and adapt. In our results, previous experience and planning created the stability to absorb and adapt, and the flexibility to continue adapting to the current flood if needed.

A driver of the stability and flexibility appeared to be the relationships between individuals and groups that delimited actions and decision-making for services during floods. Health systems are populated by health workers and community members with the power to make choices, and the social networks, relationships and collaborations among them have been identified as influencers of resilience.[13 15 26 33 34] In this study, relationships influenced how knowledge was collected and shared during floods. Being able to rely on higher levels and other providers or staff was fundamental to the interdependent preparation and response described by the participants. Understanding the boundaries of the decision-making space appears as a key component to taking action, as seen in a study of decision-making in Uganda.[35] In the context of the Cambodian health system, where decision-making is already hierarchical,[36 37] top-down leadership may have helped counteract uncertainty when responding to floods; during floods, participants focused on actions and decisions within their own responsibilities, shifting decisions beyond their remit upwards in the hierarchy. However, further work would be needed to understand and compare the effects of hierarchy on uncertainty and decision-making in non-hierarchical contexts.

The descriptions by health centre providers of their close, positive relationships with local community members contrast with their descriptions of a steady decrease in visits to health centres. An earlier study on community management of pregnancy and childbirth during floods identified a low degree of trust and sense of ownership in health centres among community members.[38] In this study, participants described trying to increase trust and demand for services year-round, regardless of floods. This could be linked to their perception of floods as a strain to the system rather than a shock. The main challenge identified during floods was not coping with or responding to health needs, but how the usual conditions of the public health system constrained the quality or quantity of care the public health system could give. Without a strong basis of trust and quality when floods were not present, there may have been fewer incentives for community members to visit public health centres during floods, when barriers to care were higher.[38] Still, the participants appeared to have a strongly anchored belief in their roles and the care they provide, and were able to build relationships with community members, which may be seeds for constructing a more legitimate system.[12 34 39 40]

Further differentiating the concepts of shocks and stresses could be useful to understand how characteristics of resilience might vary depending on what challenges a system faces.[13 41] Both seasonal and occasional floods were described as a strain on the health system—acute, external events that were also chronically repetitive in nature and aggravated existing everyday challenges—bridging current definitions of shocks and stresses.[14] This suggests that the normal pattern of floods in the districts is not a major challenge to the health services. Since floods are projected to gradually become more frequent with climate change, this is an argument for promoting the concept of everyday resilience towards repeated seasonal and occasional floods, which can in turn promote resilience to more acute, extreme shocks.[3 5]

## Study limitations

Some desirability bias is likely present among participants at higher levels of authority, such as the health departments, when it was harder to achieve depth and capture informal views than with lower level participants. Interviews with the participants who were previous students of a data collector were checked for respondent and researcher biases; there were no clear differences. Interviews that gave practical examples of participants' rationale for their actions during floods were considered higher quality in the study. Using a nascent conceptual framework with overlapping dimensions meant data had to be abstracted to a high degree during analysis. We dealt with this by using a data-driven approach during coding and then allowed the categories and themes to be influenced by the framework.

## CONCLUSIONS

Public health facilities and health departments appeared to have the capacity to absorb and adapt to seasonal and occasional floods in Cambodia. The boundaries on decisions and actions and pre-existing flood plans for facilities and departments at multiple levels created stability and flexibility when preparing and responding to floods. The apparent success of the system in responding to floods leads us to conclude that strategies that enhance stability and flexibility may foster the capacity for health system resilience. However, the impact of the floods in this study was minor compared with persistent, system-wide challenges to health service functioning. Health systems that are chronically strained by repeated external events may benefit most from health system strengthening efforts to everyday challenges that can pay off during stronger shocks.

**Acknowledgements** The authors would like to thank all the clinicians and staff who gave their time to take part in this study. We also thank the data collectors for their outstanding work, Sophea Men for her translation of the transcripts and Vannarath Te for his contributions during piloting.

**Contributors** DDS, IP, CH, HMA and JvS designed the study. DDS and DT led the data collection. HMA took part in the piloting. DDS is the guarantor, and led the data analysis and wrote the manuscript with input from all coauthors. All authors agreed on the final version of the manuscript.

**Funding** This work was supported by the Swedish Research Council (Dnr: 2016-05678).

**Competing interests** None declared.

**Patient consent for publication** Not applicable.

**Ethics approval** This study involves human participants and was approved by the National Ethics Committee for Human Research in Cambodia (NECHR, 204 and 276) and the Swedish Ethical Review Authority (Dnr: 2019-02458).

**Provenance and peer review** Not commissioned; externally peer reviewed.

**Data availability statement** No data are available. This is a qualitative study of a small, specific population in two unique geographic regions of rural Cambodia. Making the full data set publicly available could potentially breach the privacy that was promised to participants when they agreed to take part and the ethics approval granted from the National Ethics Committee for Human Research in Cambodia and the Swedish Ethical Review Authority. Therefore, the authors will not make the full transcripts available to a wider audience.

**ORCID iDs**
Dell D Saulnier http://orcid.org/0000-0001-7761-0737
Claudia Hanson http://orcid.org/0000-0001-8066-7873

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
