## [Reviewer comments · BMJ Open]

ARTICLE DETAILS

TITLE (PROVISIONAL)	'We have a plan for that': A qualitative study of health systems resilience through the perspective of health workers managing antenatal and childbirth services during floods in Cambodia
AUTHORS	Saulnier, Dell; Thol, Dawin; Por, Ir; Hanson, Claudia; Von Schreeb, Johan; Alvesson, Helle

VERSION 1 – REVIEW

REVIEWER	Wolff, Kristina The Dartmouth Institute for Health Policy & Clinical Practice, Learning Innovation Lab
REVIEW RETURNED	25-Aug-2021

GENERAL COMMENTS	The general premise of the paper is interesting. However the presentation of information is superficial, not well described or explained. This makes the paper vague and most claims unsupported or meaningful. The paper needs major revision as it isn't well developed.
--

REVIEWER	Haldane, Victoria University of Toronto Dalla Lana School of Public Health, Institute of Health Policy Management and Evaluation
REVIEW RETURNED	27-Sep-2021

GENERAL COMMENTS	This is a timely article that does the important work of teasing out factors contributing to resilience-building in health systems prone to environmental shocks. I applaud the authors for taking this lens to the study and found it interesting to read. I offer some comments for the authors to consider: Major comments: Conceptual/Analytical framework: - The manuscript could more clearly present the framework employed. I would consider making a specific section in the methods titled 'conceptual framework' or 'analytical framework' (depending on how the authors used it) or include this as a section within your 'data analysis' section. Consider moving pg 4 line 17-25 and Pg 5 line 11 to 13 here for clarity, and this section would now offer opportunity to elaborate the information on pg 7 lines 8 – 9. This will greatly help readers who are unfamiliar with the framework and the way it defines and operationalizes terms such as 'knowledge, uncertainty, interdependence, and legitimacy.' Study setting section: - This section is a bit sparse and would benefit from being a more complete picture of the setting. Suggest the authors consider
---

moving pg 5 Line 5-9 to the study settings section to better highlight and provide a concise overview of the flood related challenges, preparedness, and responses in Cambodia as well as the health system and service delivery organization points currently raised.

Methods:

- Did the interviews use an interview guide and was this guide based on a specific framework or approach? (E.g. your conceptual framework) If so please add some details here.
- The study design, participants, and recruitment section would benefit from some elaboration on what is meant by a 'heterogeneous sample' – this may include details on whether participants purposively sampled based only on location or were other factors (profession, gender, age, etc) also considered.
- Some details on how participants were approached and recruited would further strengthen this section
- Please add details on whether and how informed consent was taken
- Data collection – it would be beneficial to provide a note as to why interviews were conducted at the end of the reason season (if this time was purposively chosen) and also to specify the months when this occurred for those unfamiliar with the setting
- Data collection – were the transcripts (or a sub-sample) reviewed by anyone to ensure accuracy? Or was this a part of the quality assessment process in any way?

Results

- Table 2 could benefit from the addition of exemplary quotes in a 4th column to better illustrate the categories and sub-categories and bring together the findings.
- Figure 1 is important to the text, and I think would be better placed at the start of the results section as a way to more concretely tie together the use of the framework and the analysis and interpretation of the results. The text could then more clearly reflect these linkages.

Overall

- The discussion (pg 16, line 4-7) offers a clear conceptualization of the difference the authors make between a shock, a strain, and a stress, this should be more clearly made in the introduction as throughout strain and stress are used in a way that appears to be interchangeable, but perhaps this is not the intention. If the authors are making the distinction that they make in the discussion throughout the manuscript this point needs to be made more explicit earlier in the manuscript and reinforced throughout.

Minor comments:

Abstract:

- Missing a period at the end of the participants section.
- Results section line 16 'knowing the boundaries' is unclear, does this refer to flood boundaries? If so consider adding in.

Introduction:

- It would be clearer if the sentences around extreme weather events and floods was a separate paragraph (Page 4 Lines 8 -13)
- Page 5 Line 5 – 'has been an ongoing priority since 2008' please clarify for which organization this has been a priority (e.g., the National Committee for Disaster Management) or whether this is a more general priority in Cambodia

	Methods  - Data collection – pg 6 line 17 – the last part of this sentence is unclear, is it meant to convey that through these conversation it was determined that the interviews capture relevant information? Results  - Pg 8 line 14 – suggest to add ‘flood boundaries’ for clarity - Pg 10 line 21 – clarify what is meant by ‘higher levels’ - Pg 10 line 25 – what is OD?
--	---

VERSION 1 – AUTHOR RESPONSE

Reviewer: 1

The general premise of the paper is interesting. However the presentation of information is superficial, not well described or explained. This makes the paper vague and most claims unsupported or meaningful. The paper needs major revision as it isn't well developed.

Response: Thank you for the concerns that are raised and for the marked PDF document. We have reworked sections of the manuscript in response to the suggestions and edits included in the PDF. Replies to the PDF comments are below and changes are marked in the manuscript. We hope that we have been able to clarify the points the reviewer raised.

This point is very broad and would be particularly confusing if the reader doesn't know what organizational resilience is or anything about complex adaptive systems.

Response: We see the reviewer's point and have removed the sentence, as we agree that it does not add much information to our point about a lack of research on generating health system resilience.

This is an interesting aim. A little more detail about the Dimensions of Resilience Governance Framework in the introduction would be helpful especially if an example was provided.

Response: Thank you for the comment, which is in line with a request from Reviewer 2. We have added a subsection on the conceptual framework with additional detail (Page 5-6, Lines 19-3).

There is a lot of large ideas here but how health system resilience connects to the DRG framework as applied to antenatal and childbirth is unknown. More detailed discussion is needed in the introductory section.

Response: Thank you for the comment. We have added additional information about the rationale for antenatal and birth services in the Methods section to clarify why it is was chosen to explore experiences providing services (Page 6, Lines 15-21).

How do these facilities vary? More information is needed here. For example, a large public health facility is more likely to be able to response to a crisis than a small one. How does this varation impact or tie into your premise about preparedness or resilience?

Response: This is an interesting point. However, the health centers and hospitals we included were similarly sized and structured, following the national guidelines from the Ministry of Health (Page 7, Lines 10-11). The facilities had similar experiences in their districts during floods. We believe that this is clear in the results, where the major difference in experiences is between the levels of the system, as we have described in the sub-category ‘Health centers as the lynchpin for the health response to floods’.

Is it really a choice or is it impacted by geographic location, income, access to transportation, family or town or other types of pressures? This is important because access to care and access to appropriate care has an impact on resilience.

Response: We fully agree with the statement. The line was originally included to indicate that both options for care are available to women. However, the basis of choice for care is far broader than the statement implies. We have therefore removed it from the paper as it is beyond the scope of what we wanted to show in this study.

Population size, demographics, geography and things like available resources in the two districts should be noted as well.

Response: Please see the additions to the Methods section (Page 6, Lines 7-11 and 22-23; Table 1). We have added details per the comments from both reviewers.

More information about participants and their 'role' is necessary for context and understanding.

Response: In the method section, we have clarified that we meant role as type of job that the participant held and added additional detail (Page 7, Line 17).

It would be helpful to readers if these dimensions were introduced earlier in the paper.

Response: Thank you for the comment. The dimensions are described in the conceptual framework section in the introduction, and we have aligned the language in the methods section (Page 8, Line 13).

How participants were recruited needs to be explained and to include information such as who recruited them, was their participation voluntary, were they paid, etc.

Response: Thank you for the comment. We have added additional information on participant recruitment to the Methods (Page 7, Lines 12-15; Page 8, Lines 2-4; Page 8, Lines 9-10).

This point needs more explanation. What is a 'topic guide'? A person, a document, a participant.....?

Response: The text has been changed to clarify that the topic guide is an interview guide (Page 8, Line 12).

'the information power concept' needs explanation both in terms of what it is, how it is used and how it is meaningful here.

Response: We agree and have now included a brief description of information power and how the relevance of the information can help guide sample size, rather than the concept of saturation (Page 8, Lines 20-22).

Table should be on one page or in an appendix so the entire table can be read at once.

Response: While we agree, the manuscript has been formatted according to the journal's guidelines and we have left it in its current format.

This needs more detail.

Response: Thank you for the comment. We have added an explanation of data-driven analysis into the Data Analysis section that explains its use to derive codes from the data rather than from using theory (Page 9, Lines 6-7).

As noted previously - the framework needed more explanation. Up until this point I didn't know what the 'concept of knowledge' is, how it is used and 'who' is it used by or applied to?

Response: We see that our original explanation of the framework was insufficient and have added more detail on the framework to the introduction in response to the comments from reviewers (Page 5-6, Lines 19-3).

How the authors arrived at this theme is unknown. There's no supporting information to help the reader understand how these themes were identified.

Response: Thank you for the comment. We have clarified the analysis process in the Data Analysis section to show that codes were first derived from the data, then grouped into categories with the help of the framework, and we then identified one theme that united the categories (Page 10, Lines 4-10).

Reviewer: 2

This is a timely article that does the important work of teasing out factors contributing to resilience-building in health systems prone to environmental shocks. I applaud the authors for taking this lens to the study and found it interesting to read. I offer some comments for the authors to consider:

Response: We would like to thank the reviewer for their thoughtful and helpful review. We hope that our answers to the comments are satisfactory in responding to the reviewer's hesitations.

Major comments:

Conceptual/Analytical framework:

The manuscript could more clearly present the framework employed. I would consider making a specific section in the methods titled 'conceptual framework' or 'analytical framework' (depending on how the authors used it) or include this as a section within your 'data analysis' section. Consider moving pg 4 line 17-25 and Pg 5 line 11 to 13 here for clarity, and this section would now offer opportunity to elaborate the information on pg 7 lines 8 – 9. This will greatly help readers who are unfamiliar with the framework and the way it defines and operationalizes terms such as 'knowledge, uncertainty, interdependence, and legitimacy.'

Response: We agree with the recommendation and have chosen to add a subsection to the introduction called 'Conceptual framework' (Page 5-6, Lines 19-3). The framework was used to guide the theory of the study design and the data collection, as well informing the data analysis. We have kept the description separate from the 'data analysis' section to help keep this clear. In the framework subsection, we have also added some additional detail on the framework.

Study setting section:

This section is a bit sparse and would benefit from being a more complete picture of the setting. Suggest the authors consider moving pg 5 Line 5-9 to the study settings section to better highlight and provide a concise overview of the flood related challenges, preparedness, and responses in Cambodia as well as the health system and service delivery organization points currently raised.

Response: Thank you for this comment and we agree with the point. We have added information on health, flooding, and the health in Cambodia to the setting (Page 6, Lines 7-11 and 22-23; Table 1; Box 1). After adding the additional information, we did not move the lines from the introduction as the reviewer suggested, as we feel they are useful in the introduction to describe the rationale for the study.

Methods:

Did the interviews use an interview guide and was this guide based on a specific framework or approach? (E.g. your conceptual framework) If so please add some details here.

Response: We see that our use of the term 'topic guide' was unclear and have switched it to 'interview guide'. The guide was based on the conceptual framework and we have added a few words to make this clearer. The interview guide is also available as a supplement file (Page 8, Lines 11-14).

The study design, participants, and recruitment section would benefit from some elaboration on what is meant by a 'heterogeneous sample' – this may include details on whether participants purposively sampled based only on location or were other factors (profession, gender, age, etc) also considered.

Response: Thank you for the comment. We have added this information to Page 7, Line 17.

Some details on how participants were approached and recruited would further strengthen this section

Response: We have added additional information on how participants were approached and recruited to the Study design section (Page 7, Lines 2-4).

Please add details on whether and how informed consent was taken

Response: We see that this information was not clear and have moved it to the Data collection section (Page 8, Lines 9-11).

Data collection – it would be beneficial to provide a note as to why interviews were conducted at the end of the reason season (if this time was purposively chosen) and also to specify the months when this occurred for those unfamiliar with the setting

Response: Thank you for the comment. We have added additional information about obtaining information on the history of flooding at each site (Page 7, Lines 12-15) in conjunction with an explanation of why the end of the rainy season was chosen to ensure that we could capture relevant, recent flood experience (Page 8, Lines 8-9).

Data collection – were the transcripts (or a sub-sample) reviewed by anyone to ensure accuracy? Or was this a part of the quality assessment process in any way?

Response: Thank you for raising the point about quality. All transcripts were translated by a single translator who listened to the recordings while translating the text. Discrepancies were noted by her and discussed with the team, as well as uncertainty about any of the concepts or statements. This information has been added on Page 8, Lines 24-25.

Results

Table 2 could benefit from the addition of exemplary quotes in a 4th column to better illustrate the categories and sub-categories and bring together the findings.

Response: Thank you for the suggestion. We agree that additional quotes would help illustrate the categories and sub-categories, and have chosen to add them to the text in the results (Page 12, Lines 16-18; Page 13, Lines 19-22; Page 16, Lines 11-15)

Figure 1 is important to the text, and I think would be better placed at the start of the results section as a way to more concretely tie together the use of the framework and the analysis and interpretation of the results. The text could then more clearly reflect these linkages.

Response: Thank you for the helpful suggestion. We have moved Figure 1 to the start of the Results section to improve the connection between the framework links and the results (Page 11, Lines 7-15).

Overall

The discussion (pg 16, line 4-7) offers a clear conceptualization of the difference the authors make between a shock, a strain, and a stress, this should be more clearly made in the introduction as throughout strain and stress are used in a way that appears to be interchangeable, but perhaps this is not the intention. If the authors are making the distinction that they make in the discussion throughout the manuscript this point needs to be made more explicit earlier in the manuscript and reinforced throughout.

Response: Thank you for this comment. We have reworded a section in the introduction, including removing the word 'strain' from the description of stress. The purpose is to show that the floods observed in this study are strains, events that fell between the common definitions of stresses and shocks and will have different implications for the system. We have made this point clearer in the introduction and have changed a word in the results to avoid using stress and strain in an overlapping manner, which was our error (Page 4 Lines 17-22; Page 12, Line 2).

Minor comments:

Abstract:

Missing a period at the end of the participants section.

Response: Period added.

Results section line 16 'knowing the boundaries' is unclear, does this refer to flood boundaries? If so consider adding in.

Response: Clarified as boundaries on decision making

Introduction:

It would be clearer if the sentences around extreme weather events and floods was a separate paragraph (Page 4 Lines 8 -13)

Response: These lines are now a separate paragraph, starting on Line 9 on Page 4.

Page 5 Line 5 – 'has been an ongoing priority since 2008' please clarify for which organization this has been a priority (e.g., the National Committee for Disaster Management) or whether this is a more general priority in Cambodia

Response: Clarified as a general government priority in Cambodia (Page 5, Lines 9-10).

Methods

Data collection – pg 6 line 17 – the last part of this sentence is unclear, is it meant to convey that through these conversation it was determined that the interviews capture relevant information?

Response: We have added more detail on the concept of information power to help explain the second part of the sentence (Page 8, Lines 20-22).

Results

Pg 8 line 14 – suggest to add 'flood boundaries' for clarity

Response: We have added 'decision-making' to the sentence to clarify that decision-making are the boundaries we are interested in (Page 10, Line 24).

Pg 10 line 21 – clarify what is meant by 'higher levels'

Response: Clarified as a higher organizational levels of the health system (Page 13, Lines 24-25).

Pg 10 line 25 – what is OD?

Response: Thank you for catching the mistake, which has been fixed. (Page 14, Line 3).

VERSION 2 – REVIEW

REVIEWER	Haldane, Victoria University of Toronto Dalla Lana School of Public Health, Institute of Health Policy Management and Evaluation
REVIEW RETURNED	28-Nov-2021
GENERAL COMMENTS	The authors have addressed my prior feedback completely, I believe no further revisions are required.